# An Emerging Human Parechovirus Type 5 Causing Sepsis-Like Illness in Infants in Australia

**DOI:** 10.3390/v11100913

**Published:** 2019-10-03

**Authors:** Anthony Chamings, Kwee Chin Liew, Emily Reid, Eugene Athan, Amy Raditsis, Peter Vuillermin, Yano Yoga, Leon Caly, Julian Druce, Soren Alexandersen

**Affiliations:** 1Geelong Center for Emerging Infectious Diseases, Geelong, VI 3220, Australia; 2Deakin University, School of Medicine, Geelong, VI 3220, Australia; 3Barwon Health, University Hospital Geelong, Geelong, VI 3220, Australia; 4Australian Clinical Labs, Geelong Laboratory, Geelong, VI 3220, Australia; 5Victorian Infectious Diseases Reference Laboratory (VIDRL), Doherty Institute, Melbourne, VI 3000, Australia

**Keywords:** parechovirus, picornaviral epidemiology, recombination, genome sequencing

## Abstract

Human parechovirus (HPeV), particularly type 3 (HPeV3), is an important cause of sepsis-/meningitis-like illness in young infants. Laboratory records identified a total of ten HPeV-positive cases in Southeastern Australia between January and July 2019. The HPeV present in these cases were typed by Sanger sequencing of the partial viral capsid protein 1 (VP1) region and selected cases were further characterised by additional Sanger or Ion Torrent near-full length virus sequencing. In seven of the ten cases, an HPeV type 5 (HPeV5) was identified, and in the remaining three cases, an HPeV type 1 was identified. The HPeV5-positive cases were infants under the age of 3 months admitted to hospital with fever, rash, lethargy and/or sepsis-like clinical signs. Near full-length virus sequencing revealed that the HPeV5 was most likely a recombinant virus, with structural genes most similar to an HPeV5 from Belarus in 2018, and a polymerase gene most similar to an HPeV3 from Australia in 2013/14. While HPeV5 is not typically associated with severe clinical signs, the HPeV5 identified here may have been able to cause more severe disease in young infants through the acquisition of genes from a more virulent HPeV.

## 1. Introduction

Human parechoviruses are rapidly evolving picornaviruses that may cause sepsis-/meningitis-like illness in infants [1]. We have previously described three epidemic waves in Australia in 2013/14, 2015/16 and 2017/18 caused by a recombinant strain of human parechovirus type 3 (HPeV3) [2]. This strain of HPeV3 likely resulted from recombination of an HPeV3 Yamagata 2011 strain providing the capsid region and an unknown parechovirus providing the non-structural proteins region [3,4].

This particular strain of HPeV3 has only been recorded in Australia. However, recent studies in Europe have identified HPeV3 causing disease in young children in the United Kingdom and Germany between 2016 and 2018 with highly similar viral capsid protein 1 (VP1) sequences to the Australian recombinant HPeV3, although this is only based on partial sequencing (256–807 nucleotides) of the capsid region [5,6]. This may support our previous finding that this HPeV3 likely continued to circulate in people between Australian epidemics, based on the molecular evolution of the virus between epidemics [2]. However, in the absence of further sequencing of these European viruses, it is impossible to determine their exact HPeV3 lineage, as these viruses frequently recombine [3,7,8]. 

More complete sequencing of human parechoviruses (HPeV), globally, is important to better understand the epidemiology, to detect recombination, and to better understand the role of potential pathogenicity determinants, such as the presence/absence of an an arginine-glycine-aspartic acid (RGD) receptor motif [3,9] or changes in the non-structural proteins [2]. Although HPeV3, in general, and perhaps the Australian recombinant in particular [1,3,4], appears to cause more severe disease in infants, sequencing should not be restricted to HPeV3, as any type of parechovirus may acquire genes from a more virulent HPeV through recombination [10].

In June and July 2019, two infants aged 1 month or less presented to University Hospital Geelong with fever, rash and irritability and both were admitted for treatment. Both infants tested positive for HPeV and samples were subjected to detailed virus sequencing. Here, we describe the detection of a novel recombinant HPeV type 5 (HPeV5), the clinical presentation of the affected infants, and retrospective epidemiological analysis of HPeV-positive infants from Southeastern Australia in 2019.

## 2. Materials and Methods

### 2.1. Clinical Samples and Sample Collection

Samples from two HPeV-positive cases (G001-19/Vic/Jun/19 and G002-19/Vic/Jul/19) presented to the Geelong Hospital in Victoria, were sent to the Geelong Centre for Emerging Infectious Diseases (GCEID) for sequencing. A retrospective analysis was then conducted at the Victorian Infectious Disease Reference Laboratory (VIDRL), Melbourne, Victoria, of all human parechovirus-positive samples collected from sick infants presented to hospitals in Southeastern Australia between February and July 2019. Samples taken by clinicians for parechovirus testing included faeces, nasopharyngeal swabs and cerebrospinal fluid (CSF). Samples were included in this study under ethical exemption from the Barwon Health Research Ethics Committee (Ref No. 16/191). Consent for case descriptions was obtained from families of both infants described in detail as per Barwon Health Research Ethics Committee guidelines. Basic clinical information on all cases is presented in Table 1.

### 2.2. Nucleic Acid Extraction and cDNA Synthesis

Nucleic acid from samples submitted to VIDRL was extracted using the Qiagen Qiamp 96 kit and a Qiacube HD extraction robot, as per the manufacturer’s instructions (Qiagen, Hilden, Germany). RNA was reverse transcribed using Bioline’s Sensifast synthesis kit (Bioline, UK), as per kit instructions, and cDNA was stored at −20 °C until processing. This cDNA was used in the parechovirus detection, typing and sequencing PCRs described below.

Samples sent to GCEID for next-generation sequencing, were first processed to enrich virus particles, as previously described [11,12]. Briefly, nasopharyngeal swabs were placed in 1 mL of universal transport media (UTM) (Copan, Italy), briefly homogenized in a TissueLyzer (Qiagen, Hilden, Germany) at 25 Hz for 2 min, and then centrifuged at 17,000× *g* for 3 min. A quantity of 500 µL of supernatant was then spun through a 0.8 µm filter spin column at 2000× *g* for 2 min (Vivaclear mini 0.8 µm PES filters, Sartorius). A quantity of 450 µL of filtrate was then ultracentrifuged at 178,000× *g* for 1 h at room temperature. The resulting pellet was resuspended in 130 µL of PBS, and treated with nucleases for 2 h, as previously described [11]. The virus suspension was then treated with 200 µM PMAxx dye (Biotium) for 10 min in the dark, before being exposed to blue light for 30 min. Nucleic acid was extracted using the QIAamp Viral RNA Mini Kit (Qiagen, Hilden Germany) spin column, as per the manufacturer’s instructions. Nucleic acids were also extracted directly from samples without prior virus enrichment using the same kit for PCR and Sanger sequencing.

### 2.3. Next-Generation Library Preparation and Sequencing

Reverse transcription and amplification were performed using the Seqplex RNA Amplification Kit (Sigma), as per the manufacturer’s instructions, and the amplified DNA quantified using an Agilent 2100 Bioanalyzer and high-sensitivity DNA chips (Agilent, Germany). The barcoded libraries were prepared with the Ion Plus Fragment Library Kit (Thermofisher Scientific), using the IonXpress barcodes, as per the manufacturer’s instructions. Final library quantification was performed using the Ion Library TaqMan™ Quantitation Kit (Thermofisher Scientific).

Barcoded libraries were loaded on to an Ion 530 chip using the Ion Chef templating robot, and sequenced using an Ion Torrent S5 XL sequencer (Thermofisher Scientific). Human parechovirus and rhinovirus sequences were obtained by mapping the obtained sequencing reads to HPeV and rhinovirus reference genomes using MagicBLAST (NCBI). The consensus sequences from the mapping were obtained from the Integrative Genomics Viewer (IGV) (Broad Institute, University of California) and assembled in Geneious 11.1.5 (Biomatters, New Zealand).

### 2.4. PCR and SANGER Sequencing

Initial detection of parechovirus in the clinical samples was performed at VIDRL using a 5’ UTR PCR, as described previously [3]. Samples not sequenced by the Ion Torrent were typed using a VP3-VP1 PCR [13] at VIDRL. Additional sequence was obtained from two HPeV5-positive samples using a 3D polymerase PCR [7] and an in-house nested HPeV 2C-3A PCR. This 2C-3A PCR used 2 µL of cDNA in a 10 µL reaction using 1× Amplitaq Gold 360 master mix (Life Technologies), 2 µM Syto9 and 1 µM of each primer Parecho-AC-4338-4360F (ACNGCHAGYGARTTYATGGAYGG) and primer Parecho-AC-5103-5130R (ATACTGGCACTRGTRACAGCTGA). The nested 2C-3A PCR took 1 µL of the first 2C-3A PCR reaction in a 10 µL reaction with 1× Amplitaq Gold 360 master mix, 2 µM Syto9 and 1µM of each primer Parecho-AC-4578-4609F (AARAGAAGATTCCCATACATYATGCACATTAG) and primer Parecho-AC-4997-4775R (TCGTGTCTGGYTTGCAACTTTRG). Both PCRs were incubated at 95 °C for 10 min, then 35 cycles of 95 °C for 30 s, 55 °C for 30 s and 72 °C for 45 s, with a final incubation of 72 °C for 3 min.

PCR products were purified in a 2% Size-Select E-Gel (Life Technologies) and sequenced on a Hitachi 3500 genetic sequencer (Thermofisher) using the BigDye Terminator v3.1 Cycle Sequencing kit (Life Technologies), as per the manufacturer’s recommendations. 

### 2.5. Recombination and Phylogenetic Analysis

Possible recombination was investigated by comparing the HPeV5 genome obtained from G001-19 against the two nearest complete HPeV3 and HPeV5 sequences from GenBank: AM23749.1 HPeV5 T92-15 detected in California in 1992 and KY55671.1 HPeV3 CSF19, detected from Australia in 2013 in Simplot [14] using a window size of 200 and step size of 20. 

Phylogenetic analysis was performed by downloading HPeV5 VP1 sequences from GenBank (National Centre for Biotechnology information (NCBI), USA) in August 2019, and aligning the sequences using Muscle [15] in Geneious (Biomatters, New Zealand). Sequences with ambiguous bases, or those which were truncated, were removed from the alignment. The optimum model for transitions/transversions was determined in Mega X [16], and the maximum-likelihood phylogenetic tree was generated in Mega X using 1000 bootstrap replicates. The resulting tree was visualized in Figtree 1.4.4 [17].

## 3. Results

### 3.1. Clinical Presentation of the HPeV-Positive Infants

Retrospective analysis of laboratory records identified a total of 10 positive HPeV cases from Southeastern Australia submitted to VIDRL between January and July 2019. Basic clinical information from nine of the cases is available (Table 1), and detailed clinical information is available for the two cases presented below. Nine cases were from Victoria and one from Tasmania. Seven cases were in infants aged 3 months or less. 

The case-G001-19/Vic/Jun/19 infant presented at 2 weeks of age in late June 2019 with a 24 h history of irritability, rash and fever. There was no significant past history, having been born uneventfully at term gestation. The infant was tachycardic (heart rate of 180 beats per minute (BPM)) and had impaired perfusion with delayed capillary refill. An erythematous macular rash was apparent across the torso. The fontanelle was full but soft. Haemoglobin was 161 g/L, white cell count was 5.7 × 10^9^/L, lymphocytes 1.4 × 10^9^/L, neutrophils 3.5 × 10^9^/L and platelets 153 × 10^9^/L. C-reactive protein levels were not elevated, and the general chemistry was unremarkable. Cerebrospinal fluid (CSF) was blood-stained with a red cell count of 8829 × 10^6^/L and a white cell count of 4 × 10^6^/L. The protein content was slightly elevated at 1.06 g/L and the glucose was normal. CSF, blood and urine bacterial cultures were negative. CSF was positive for parechovirus by PCR and negative for herpes simplex virus and enterovirus. A nasopharyngeal (NPA) swab tested negative for influenza A/B virus by rapid test and a second NPA swab tested positive (PCR cycle threshold (Ct) value of 27.6) for rhinovirus by a multiplex PCR assay for respiratory pathogens. The infant was treated empirically with intravenous antibiotics (cefotaxime and benzyl-penicillin) for 48 h and acyclovir for 24 h along with intravenous (IV) saline and dextrose for circulatory support and hydration. The infant improved clinically and was discharged after 3 days, but continued to have some irritability for the following week.

The case-G002-19/Vic/Jul/19 infant presented in late July 2019 at one month of age after a one-week history of nasal congestion and then sudden deterioration overnight with fever, poor feeding and irritability. This infant also had loose stools. There was no significant past history, having been born at term from an uneventful pregnancy. The infant presented mottled, tachycardic (191 BPM), tachypnoeic (60 breaths/min) and febrile (38.5 °C) and was noted to have episodes of oxygen saturation to 80%. There was a blanching macular rash on the torso. The fontanelle was soft. Haemoglobin was 107 g/L, white cell count 6.5 × 10^9^/L, neutrophils 3.5 × 10^9^/L, lymphocytes 3.1 × 10^9^/L and platelets 355 × 10^9^/L. Neutrophil count on repeat specimen during the illness became mildly neutropenic at 1.2 × 10^9^/L. C-reactive protein levels were mildly elevated at 7.2 mg/L. General chemistry was unremarkable. Cerebrospinal fluid was bloodstained with a red blood cell count of 403 × 10^6^/L and white cell count of 22 × 10^6^/L. The protein content was unremarkable at 0.33 g/L and the glucose level was normal. CSF and blood cultures were negative (obtained post empiric antibiotic administration). Blood cultures were contaminated with Staph. Epidermidis. An NPA swab was borderline positive/negative for rhinovirus by PCR (Ct value of 37.8) and negative for influenza A/B virus. PCR on CSF was positive for parechovirus and negative for enterovirus and herpes simplex virus. The chest X-ray was normal. The infant was treated empirically with intravenous antibiotics (flucloxacillin and cefotaxime) for 48 h and was also treated with intravenous saline and dextrose for circulatory support and hydration. The infant improved clinically and was discharged after 3 days.

### 3.2. Detection of A Novel HPeV5 in Samples from the Infants

Immediately after PCR detection of HPeV infection in G001-19, a nasal swab, plasma and CSF collected from the infant during their hospitalization were sent to GCEID and PCR-amplified and sequenced using PCRs targeting VP1, 2C and 3D. All samples were positive, and the sequences obtained for each sample were identical in the respective PCRs (accession numbers MN193506-MN193511). The VP1 sequencing identified HPeV5 in the samples. Initial Sanger sequencing was followed by Ion Torrent sequencing of the G001-19 nasal swab in which a 7268 nt-long sequence of an HPeV5 was obtained (GenBank accession: MN212905). This consisted of 620 nt of the 5’UTR, the full polyprotein (6561 nt), the complete 87 nt of the 3’UTR and part of the poly-A tail. An RGD motif was present at the 3’ end of the VP1 sequence. The genome was assembled from 29,031 reads or 0.35% of the total reads from this sample.

For case G002-19, a nasal swab was sent to GCEID for sequencing upon detection of an HPeV in the infant’s CSF. A 7136 nt HPeV5 sequence was obtained from the NPA swab of case G002-19. This was made up of 488 nt of the 5’UTR, the full polyprotein and 3’UTR and part of the poly-A tail (GenBank accession: MN451649). The genome was assembled from 4288 reads or 0.08% of the total reads generated from this sample. The HPeV5 from G002-19 was 99.96% (7136 of 7139 nucleotides) similar to the G001-19 virus in the nucleotide sequence. The polyproteins of these viruses were identical in terms of their amino acid sequence.

The nearest complete HPeV sequence in GenBank to the G001-19 HPeV5 was HPeV5 strain T92-15, detected in California in 1992 (GenBank accession AM235749.1). When individual virus proteins were compared by Basic Local Alignment Search Tool (BLAST (NCBI)) to the sequences in Genbank, it was apparent that different genes shared high homology with different HPeV sequences in GenBank. The VP1 was 99.85% similar to HPeV5 detected in Belarus (HPeV_5_19393_2018_BLR, GenBank accession: MK168005.1) and 99.71% to HPeV5 detected in Germany in 2018 (BE.2211/HE/DE/2018, GenBank accession: MK291280.1), protein 2C was 90.58% similar to an HPeV1 31170176 (LC318432) from Japan, 2017 and the polymerase (protein 3D) was 98.37%–98.58% similar to HPeV3 (FEC22-KY556674.1, FEC21-KY556673.1 and CSF19-KY556671.1) detected in Australia during 2013 and 2014. 

NPA swabs from both cases were reported positive for rhinovirus by real-time PCR. The rhinovirus PCR for case G001-19 had a Ct of 27.6; however, case G002-19 was positive at a very high Ct of 37.8 (borderline positive/negative). A full-length rhinovirus B sequence was obtained from G001-19 by sequencing (GenBank accession MN212904.1). This sequence was generated from 156437 reads (1.86% of the total reads). No rhinovirus reads were seen in G002-19, which suggested that the detection by PCR may have been of a very low level of residual virus not actively replicating in this patient at the time of sampling. This would be consistent with this infant having a history of nasal congestion a week prior to presentation.

### 3.3. The HPeV5 Likely A Product of Multiple Recombination Events between Different HPeV Types

Simplot analysis of the G001-19 HPeV5 against reference HPeV5 (AM235749.1) and HPeV3 (KY556671.1) genomes is shown in Figure 1. The structural genes (VP0, VP3 and VP1) were most similar to HPeV5, while the non-structural genes were more similar to HPeV3. The polymerase (viral protein 3D) was highly similar (>95%) to the Australian recombinant HPeV3, which has been the predominant HPeV lineage in Australia since 2013. In particular, it was most similar to the HPeV3s sequenced from the earliest epidemic in 2013/14 (GenBank accessions: KY556671.1, KY556674.1 and KY55673.1). The middle region of the genome, which included the non-structural viral proteins 2A, 2B, 2C, 3A and 3B, only shared 85%–95% homology to either HPeV3 or HPeV5. Querying this region in BLAST found that the most similar sequence was an HPeV1 from the Netherlands from 1993 (K150-93 (GQ183023.1)), with an overall similarity of 87.8%. However, many HPeV1, 3 and 4 sequences also shared >85% similarity with this region, and, therefore, it is possible that this section of the genome was acquired from a yet to be sequenced HPeV belonging to any one of these types, and possibly even the progenitor virus for the 3’ end of the Australian recombinant HPeV3, given the similarity to the HPeV3s from the earliest Australian epidemic. The Simplot analysis indicated that the potential sites of recombination events were in the 5’ end of the 2A protein and 3’ end of the 3C protein (Figure 1).

### 3.4. Retrospective Analysis of HPeV Detected in Southeastern Australia in 2019

Retrospective analysis of all HPeV-positive cases from Southeastern Australia in 2019 found that seven of the ten cases were positive for HPeV5 based on VP1 sequencing, and three positive for HPeV1. In four HPeV5 infections, the infant had been treated for suspected meningitis or sepsis, and in 6 out of the 7 cases, CSF had been submitted for parechovirus and enterovirus testing, indicating significant clinical concern for the infant. All HPeV5-infected infants were less than 3 months of age (mean age of 1 month), while all three HPeV1-positive infants were 9 months or older and CSF had not been submitted for testing (see Table 1). The VP1 nucleotide sequences of the seven HPeV5s were 99.1% similar or higher. 

Sufficient sample remained from two of these HPeV5-positive cases (V1/Vic/Feb/19 (V1) and V5/Tas/Mar/19 (V5)) to perform additional VP1, partial 2C-3A and 3D PCRs and Sanger sequencing (accession numbers MN451643-MN451648). A 296 nt section of the 2C-3A region of the HPeV5s (corresponding to nucleotides 4589 to 4885 of G001-19) was able to be aligned from all four HPeV5’s sequenced. The HPeV5 from V1 (Feb/19) was 96.96% similar to G001-19, while the HPeV5 from V5 (Mar/19) was 97.64% similar to G001-19. V1 was 98.65% similar to V5. In contrast to the recent Australian recombinant HPeV3 sequences, these HPeV5’s possessed a glutamine (Q) residue at the amino acid residue immediately upstream to the 2C-3A cleavage motif QLENQ. V1 (Feb/19), however, had changed a glutamine for an arginine, corresponding to the residue 12 amino acids upstream of the QLENQ motif in the G001-19 polyprotein. A 607nt region of the 3D polymerase (corresponding to nucleotides 6489 to 7096) was aligned. The V1/Vic/Feb/19 HPeV5 was 98.68% similar to G001-19 and V5/Tas/Mar/19 HPeV5 was 99.01% similar to G001-19. V1 was 99.01% similar to V5. In both PCR regions, G002-19 was identical to G001-19.

At the nucleotide level, the sequenced VP1 region contained six variable sites out of 566 nucleotides (1.06%), and zero out of 188 variable amino acids sites. The 2C-3A region contained 10 variable nucleotides out of 297 (3.37%), and four variable amino acids out of 98 (4.1%). The 3D contained 10 variable nucleotides out of 608 (1.64%), and one variable out of 202 amino acids (0.5%). The difference in the rate of variability between the VP1 and 3D in both nucleotides and amino acids was not statistically significant (*p* = 0.14 and *p* = 0.52, Fisher’s Exact Test). However, the rate of variability in the 2C-3A region was significantly different to the 3D and VP1 regions in both nucleotide and amino acids (*p* = 0.017 and *p* = 0.006, Fisher’s Exact Test), suggesting that this was a hotspot of variability in this HPeV5.

### 3.5. The Parent HPeV5 Virus Belongs to A European HPeV5 Lineage

At the time of writing (August 2019), only nine full-length HPeV5 genomes were available on GenBank. However, 76 near-complete HPeV5 VP1 sequences were available, and these were downloaded and aligned with the 696nt VP1 sequences of G001-19 and G002-19. After excluding any sequences with ambiguous bases, or those which were significantly truncated at either the 5’ or 3’ end of the gene (which included the partial VP1 from cases V1 and V5), a final alignment of 567 nucleotides from 72 HPeV5 VP1 sequences was created, and a corresponding maximum-likelihood phylogenetic tree generated (Appendix A). The 2019 Australian HPeV5 VP1 sequences ranged in nucleotide similarity between 77% and 100% with international HPeV5s, but were only 80.78%–80.95% similar to HPeV5’s detected in Queensland, Australia, in 2012 and 2013 (GenBank accessions: MH931514.1 and MG712788.1) [18,19], and most closely related (92%–100% similarity) to a lineage of HPeV5 viruses detected in several countries across Europe between 2004 and 2018. Of these, the recombinant HPeV5 VP1 was most similar to the most recent sequences reported in 2018 from Germany (all >99.5% similarity) and Belarus (100% similar to G002-19) [6] (Figure 2).

## 4. Discussion

A recombinant human parechovirus type 3 has caused three epidemics in Australia approximately every two years between 2013 and 2018 [1,2,3,20]. While 2019 was predicted to be the year in which the next HPeV3 epidemic in Australia was likely to occur, instead a recombinant HPeV5 has been the predominant parechovirus infection detected in sick infants in 2019. It is unclear at the time of writing, whether these cases represent the early start of a new epidemic of HPeV infections in Australia, or whether the HPeV5 will be replaced by the Australian recombinant HPeV3 as the predominant HPeV type later this year, as the southern hemisphere Spring–Summer is when HPeV3 epidemics have been typically observed in Australia [1,2,4,18]. 

While HPeV5 is not typically associated with the more severe manifestations of parechovirus infections, such as meningitis and sepsis-like symptoms in children, the recombinant HPeV5 described here has had somewhat similar clinical manifestations to the Australian recombinant HPeV3 [1,21]. All seven HPeV5 infections in 2019 were detected in young infants less than 3 months old, and the rate of sepsis- and meningitis-like symptoms (57%) in cases was similar to that seen in infections with the Australian recombinant HPeV3 (52%) in the 2017/18 epidemic [2]. However, the clinicians at University Hospital Geelong have indicated that the two 2019 cases treated there, although severe, were not as severe as cases treated during the Geelong 2015 HPeV3 outbreak. It is possible that the acquisition of a HPeV polymerase very similar (>98% similarity) to the virulent recombinant Australian HPeV3 could result in more efficient virus replication and, therefore, more severe clinical signs in this HPeV5, resulting in the more HPeV3-like clinical manifestation. Similar intertypic recombination, and possible acquisition of HPeV3 genes, has been observed in HPeV4 isolated from infants with sepsis-like illness [10,22]. 

It has been hypothesized that HPeV3 could cause more severe disease due to the fact that it lacks the RGD integrin-binding motif and, therefore, perhaps infects a wider range of cells in the host by utilizing a cellular receptor different to other HPeV types [3,9,23,24,25,26]. However, the recombinant HPeV5 detected here possessed the RGD motif and had a high rate of meningitis-/sepsis-like symptoms reported in the infected children, suggesting that the severity of clinical disease is not dependent on the absence of this motif. We previously speculated that changes of a glutamine (Q) to an arginine (R) immediately upstream to the cleavage site of the 2C-3A proteins may be important in determining virulence [2]. While, the 2C-3A junction of the sequenced HPeV5’s in this study all possessed a glutamine at this position, this region was again a hotspot of variability with more nucleotide and amino acid changes compared to the other sequenced regions in the four sequenced HPeV5’s. Three of the four amino acid changes from the junction region occurred in 2C. Studies have shown that the 2C co-localizes with viral RNA in HPeV1-infected cells [27] and for poliovirus, another picornavirus studied in much more detail at the molecular level, direct interaction/adaptation has been demonstrated for capsid and 2C amino acid sequences [28,29]. It may, therefore, be likely that for a recently formed recombinant HPeV, the 2C must change to better bind to the new viral RNA and capsid structure. It may also be that this variability is the result of the viruses sequenced in this study represent multiple introductions of related viruses into Australia. These may have arisen when the two or more parent viruses were co-circulating in more population-dense regions overseas, resulting in multiple recombination events producing slightly different progeny viruses, which were then brought to Australia by travelers.

While no full-length sequences similar to either of the Australian recombinant HPeV3 or HPeV5 have been published from other countries at the time of writing, recent studies from Europe have indicated that, based on partial VP1 sequences, viruses closely related, at least in VP1, to the Australian HPeV5 were circulating in Europe as recently as 2018 [6]. Interestingly, the same report observed a simultaneous spike in HPeV3 infections, with several HPeV3 lineages detected, including HPeV3’s with VP1 sequences that were highly similar (≥99% similarity by BLASTn) to the Australian recombinant HPeV3s from 2013–2014 (CSF19, FEC21, FEC22 and FEC23) [4]. Europe may have, therefore, been the geographical region where the progenitor HPeV5 and HPeV3 viruses had the opportunity to recombine and produced the virus described here. However, this is purely speculative without the complete genomes of the viruses detected there in 2018. We propose that more complete genome sequencing of HPeVs should be undertaken in future surveillance to better understand the epidemiology, virulence factors and the rate of recombination of this virus.

Both of the G001-19 and G002-19 cases, described in detail here, were PCR positive for rhinovirus, although the Ion Torrent sequencing was only able to detect a rhinovirus B from patient G001-19. The lack of rhinovirus reads from G002-19 coupled with the high PCR Ct value, suggests that the PCR detection of rhinovirus in this patient may not have been of actively replicating virus, unlike in patient G001-19, but more consistent with the nasal congestion observed one week prior in this infant.

Co-infection of rhinovirus and parechovirus was not noted in any of the other cases, but rhinovirus and parechovirus testing are usually not part of the same diagnostic screens and, therefore, these coinfections are only likely to be detected in patients screened for both respiratory disease and viral meningitis, or those in which metagenomics are performed. Nevertheless, although speculative at this stage, we cannot exclude that co-infection with parechovirus and rhinovirus may contribute to more severe illness in young infants as both have a propensity to infect basal cells in the human respiratory tract [30,31], and infection with one may enable the second to more readily infect, replicate and/or spread in a host.

## Figures and Tables

**Figure 1 viruses-11-00913-f001:**
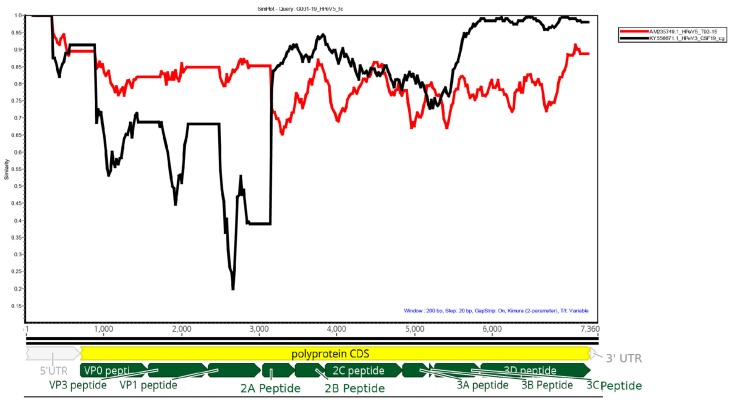
SimPlot analysis was used to plot the degree of similarity between the HPeV5 G001-19 sequence and those of the closest near-full length sequences of HPeV5 (AM235749 [16]) and HPeV3 (KY556671 [4]) available in NCBI. The position of the 5’UTR, viral peptides and 3’UTR are shown below the plot.

**Figure 2 viruses-11-00913-f002:**
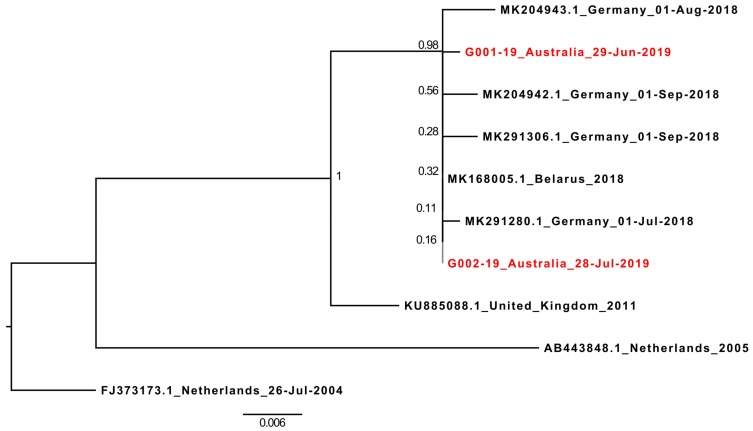
Maximum-likelihood tree of 567nt of all HPeV5 VP1 sequences from the NCBI GenBank with >90% similarity to G001-19. The tree was generated in Mega X using the GTR+G+I model. Branch support was determined with 1000 bootstrap replicates, and is shown at each node. The near full-length 2019 recombinant HPeV5 sequences described here are in red.

**Table 1 viruses-11-00913-t001:** Clinical details of the human parechovirus (HPeV)-positive cases between February and July 2019 in Southeastern Australia.

Case Number	HPeV Type	HPeV Positive Samples	Clinical Summary	Infant Age (Days)
V1/Vic/Feb/19	HPeV-5	CSF, Faeces	Suspect meningitis, fever	14
V2/Vic/Feb/19	HPeV-5	CSF	Meningitis/sepsis-like	68
V3/Vic/Mar/19	HPeV-1	Faeces	None provided	267
V4/Vic/Mar/19	HPeV-5	CSF	Lethargy and poor feeding	50
V5/Tas/Mar/19	HPeV-5	Plasma, Rectal swab	Rash	24
V6/Vic/Apr/19	HPeV-5	CSF	Fever and nasal congestion	16
V7/Vic/May/19	HPeV-1	Bowel contents	Viral illness and death	524
V8/Vic/May/19	HPeV-1 (and enterovirus)	Faeces, Rectal Swab	Vomiting, distended abdomen	615
G001-19/Vic/Jun/19	HPeV-5	CSF, Plasma, Nasal swab	Meningitis/sepsis-like, fever, rash and irritability	17
G002-19/Vic/Jul/19	HPeV-5	CSF, Nasal Swab	Meningitis/sepsis-like, fever, rash and irritability	32

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
