# Peer review of "An Emerging Human Parechovirus Type 5 Causing Sepsis-Like Illness in Infants in Australia"

_viruses, 2019, doi:10.3390/v11100913_

Round 1

Reviewer 1 Report

I have read this very interesting paper. It has a very good mix of clinic and basic applied virology.

It is a very good reflex of the questions that pediatrician and ID specialist make when have to face this new sepsis-like pictures at the hospital bases.

It is well written and easy to read, the authors highlight the importance of share the epidemiological information about this new parechoviruses, and they describe very well the methodology they used.

It has a public health impact too since all these children receive antibiotics and to have a rapid diagnosis of the correct etiology  will allow to early discharge of the patients

Author Response

Response 1: We are very pleased to hear that Reviewer 1 find our paper to be interesting, well written and a good mix of clinical and basic applied research.

Reviewer 2 Report

Although HPeV5 is not typically associated with the severe manifestations, the authors found the recombinant HPeV5, with structural genes similar to an HPeV5 and a polymerase gene similar to an HPeV3, causes the HPeV3-like severe disease in young infants in 2019 in Australia.   This article shows important findings and gives us a strong suggestion between their recombination events and pathogenicity.   My comments are as follows.

How many for “ALL HPeV positive cases” (Line 131) ?.   “All HPeV5 infected infants were less than 3 months of age, while all three HPeV1 positive infants were 9 months or older” (Lines 234-235) is an interesting finding.   Were there no other different findings than age distribution between the former group and the latter?   Meningitis/sepsis-like symptom was more commonly observed in the former group (Table 1)?   Were there no recombination events for HPeV1 strains? Don’t you include what will be going on in the next several month in this paper (it is unclear whether the HPeV5 will be replaced by the Australian recombinant HPeV3 as the predominant HPeV type later this year; Lines 284-287) ?

Author Response

Although HPeV5 is not typically associated with the severe manifestations, the authors found the recombinant HPeV5, with structural genes similar to an HPeV5 and a polymerase gene similar to an HPeV3, causes the HPeV3-like severe disease in young infants in 2019 in Australia.   This article shows important findings and gives us a strong suggestion between their recombination events and pathogenicity.   My comments are as follows.

Point 1: How many for “ALL HPeV positive cases” (Line 131)?

Response 1: Although we subsequently in that sentence point out that the number of cases is 10, we have realised that the sentence is a bit difficult to read and consequently have changed it in the revised manuscript to read “Retrospective analysis of laboratory records identified a total of 10 HPeV positive cases from south eastern Australia submitted to VIDRL between January and July 2019”.

Point 2: “All HPeV5 infected infants were less than 3 months of age, while all three HPeV1 positive infants were 9 months or older” (Lines 234-235) is an interesting finding.   Were there no other different findings than age distribution between the former group and the latter?   Meningitis/sepsis-like symptom was more commonly observed in the former group (Table 1)?

Response 2: In line 230-237 of the manuscript we describe what we know in regards to differences between the HPeV5 and the HPeV1 cases. In order to make it easier to see the details, we have added “see Table 1” to this section, although this Table is mentioned earlier in the manuscript in line 67 and line 133.

Point 3: Were there no recombination events for HPeV1 strains?

Response 3: For the three HPeV1 cases mentioned in the manuscript we do only have partial VP1 sequences sufficient to type them as HPeV1. As these cases where not meningitis-/sepsis-like illness in very young babies, they were not the focus of our investigation and apart from the typing by partial VP1 sequencing, we have not attempted any additional sequencing and can thus not say anything about recombination or not.

Point 4: Don’t you include what will be going on in the next several month in this paper (it is unclear whether the HPeV5 will be replaced by the Australian recombinant HPeV3 as the predominant HPeV type later this year; Lines 284-287)?

Response 4: Nothing significant has happened in regards to HPeV3 since submission of this manuscript and as we only expect a potential outbreak of HPeV3 to occur later in the year or perhaps early next year, if at all, we wanted to get the information about a severe strain of HPeV5 out as soon as possible to make the information available to clinicians and laboratories worldwide. If we see another outbreak of HPeV3, we will of course follow that up with a subsequent submission when and if we have results available.